# Transfer of Cardiac Mitochondria Improves the Therapeutic Efficacy of Mesenchymal Stem Cells in a Preclinical Model of Ischemic Heart Disease

**DOI:** 10.3390/cells12040582

**Published:** 2023-02-11

**Authors:** Marie-Luce Vignais, Jennyfer Levoux, Pierre Sicard, Khattar Khattar, Catherine Lozza, Marianne Gervais, Safia Mezhoud, Jean Nakhle, Frederic Relaix, Onnik Agbulut, Jeremy Fauconnier, Anne-Marie Rodriguez

**Affiliations:** 1Institut de Génomique Fonctionnelle, University Montpellier, CNRS, INSERM, 34094 Montpellier, France; 2Université Paris-Est Créteil, INSERM, IMRB, 94010 Créteil, France; 3Sorbonne Université, Institut de Biologie Paris-Seine (IBPS), CNRS UMR 8256, INSERM U1164, Biological Adaptation and Ageing, 75005 Paris, France; 4PhyMedExp, Inserm, CNRS, University of Montpellier, 34295 Montpellier, France; 5École Nationale Vétérinaire d’Alfort, IMRB, 94700 Maisons-Alfort, France; 6APHP, Hôpitaux Universitaires Henri Mondor & Centre de Référence des Maladies Neuromusculaires GNMH, 94000 Créteil, France

**Keywords:** mesenchymal stem cells, mitochondria transfer, cell therapy, metabolism, post-ischemic heart failure

## Abstract

Background: The use of mesenchymal stem cells (MSCs) appears to be a promising therapeutic approach for cardiac repair after myocardial infarction. However, clinical trials have revealed the need to improve their therapeutic efficacy. Recent evidence demonstrated that mitochondria undergo spontaneous transfer from damaged cells to MSCs, resulting in the activation of the cytoprotective and pro-angiogenic functions of recipient MSCs. Based on these observations, we investigated whether the preconditioning of MSCs with mitochondria could optimize their therapeutic potential for ischemic heart disease. Methods: Human MSCs were exposed to mitochondria isolated from human fetal cardiomyocytes. After 24 h, the effects of mitochondria preconditioning on the MSCs’ function were analyzed both in vitro and in vivo. Results: We found that cardiac mitochondria-preconditioning improved the proliferation and repair properties of MSCs in vitro. Mechanistically, cardiac mitochondria mediate their stimulatory effects through the production of reactive oxygen species, which trigger their own degradation in recipient MSCs. These effects were further confirmed in vivo, as the mitochondria preconditioning of MSCs potentiated their therapeutic efficacy on cardiac function following their engraftment into infarcted mouse hearts. Conclusions: The preconditioning of MSCs with the artificial transfer of cardiac mitochondria appears to be promising strategy to improve the efficacy of MSC-based cell therapy in ischemic heart disease.

## 1. Introduction

Ischemic heart diseases constitute a major public health issue affecting millions of people worldwide, causing high mortality and physical disabilities, as well as considerable care costs [1]. The treatment of myocardial ischemia (MI) is challenging because standard therapies have shown limited efficacy in repairing the ischemic heart. Accordingly, numerous preclinical studies have reported on the beneficial effects of cell transplantation using mesenchymal stem cells (MSCs) after MI [2,3,4]. Notably, MSC-based therapy has been shown to decrease inflammation and scar formation and improve angiogenesis and cardiac function in infarcted rodent hearts [5,6]. Despite these encouraging results, clinical trials using MSC engraftment have failed to confirm the substantial cardiac functional recovery observed in animal models, thus emphasizing the need to optimize the therapeutic efficacy of MSCs [5,6].

Among the molecular mechanisms triggering MSCs’ regenerative capacities, the surrounding microenvironment plays a critical role in governing the behavior of both resident and engrafted stem cells [7,8,9]. In particular, the stress signals released by damaged cells were found to stimulate the healing capacities of MSCs [10,11]. The recent discovery of mitochondrial translocation from micro-environmental cells to MSCs has opened up a novel paradigm whereby mitochondria may be one of the cues that is able to trigger an adaptive response in MSCs [12]. In particular, the transfer of mitochondria originating from differentiated cells or platelets modifies several properties of MSCs, specifically their proliferation [13], plasticity [14,15], cytoprotective functions [16], and pro-angiogenic potential [17].

Mitochondria play a prominent role in controlling the function and fate of a wide range of cells, including MSCs, by adapting their metabolism such as through oxidative phosphorylation and glycolysis activities to meet a cell’s needs for ATP and building blocks [18,19]. In particular, processes involved in mitochondria dynamics and remodeling, such as mitophagy, have been shown to control cell metabolism and functions. For example, mitophagy has been reported to regulate survival, proliferation and differentiation processes in several cell types including retinal ganglion cells and macrophages, as well as cancer, embryonic, and induced pluripotent stem cells [20,21,22,23].

Interestingly, the internalization of mitochondria originating from other cells has been found to stimulate MSCs’ repair properties through a process involving mitophagy activation [16]. More specifically, we previously reported that mitochondria conveyed from damaged cardiac or endothelial cells to MSCs needed to be degraded by the recipient MSCs to stimulate their cytoprotective potential [16]. 

Based on these observations, we asked whether the artificial transfer of mitochondria could improve the therapeutic potential of MSCs for ischemic cardiac repair. We found that the preconditioning of MSCs with mitochondria isolated from human cardiomyocytes improved their regenerative functions through a mechanism involving reactive oxygen species (ROS) production and mitophagy. We further demonstrated that mitochondria-preconditioned MSCs have a greater therapeutic potential than untreated MSCs when engrafted into infarcted mouse hearts. Overall, our findings offer a new paradigm towards the development of innovative cell-based therapy for ischemic heart disease.

## 2. Materials and Methods

### 2.1. Cell Cultures

All experiments were conducted using human multipotent adipose-derived stem (hMADS) cells as a model of human MSCs. The hMADS cells were isolated from the adipose tissues of young donors after the informed consent of their parents and considered surgical scraps by the Comité de Protection des Personnes, Centre Hospitalier Universitaire de Créteil in 2007 [24]. 

The human fetal ventricular cardiomyocyte RL14 cell line was purchased from the American Type Cell Culture (ATCC, LGC Standards S.a.r.l., Molsheim, France). RL14 cells were grown in DMEM/F-12 (Thermo Scientific, Waltham, MA, USA Cat# 31331028) supplemented with 10% heat-inactivated FBS, 100 U/mL of penicillin, 100 µg/mL of streptomycin, and 10 mM of HEPES [25].

The hMADS cells were expanded in Dulbecco’s modified Eagle’s medium (Thermo Scientific, Waltham, Massachusetts, USA Cat#21885025) supplemented with 1 g/L of glucose, 10% heat-inactivated fetal bovine serum (FBS) (Dominique Dutscher, Bernolsheim, France, Cat#S1810-500), 100 U/mL of penicillin, 100 μg/mL of streptomycin, and 10 mM of HEPES (Gibco, Cat#15630056). As described earlier [24], hMADS cells exhibited the following phenotypes: CD44^+^, CD49b^+^, CD105^+^, CD90^+^, CD13^+^, Stro-1^−^, CD34^−^, CD15^−^, CD117^−^, Flk-1^−^, Gly-A^−^, CD133^−^, HLA-DR^−^ and HLA-I^low^. 

Experiments were conducted with hMADS cells isolated from two young donors at passages from 10 to 30.

Interventional studies involving animals or humans and other studies that require ethical approval must list the authority that provided approval and the corresponding ethical approval code.

### 2.2. Cardiac Mitochondria Isolation and Transfer to MSCs

Cardiac mitochondria were isolated from RL14 cells using the Mitochondria Isolation Kit (Thermo Scientific, Waltham, MA, USA, Cat#89874) according to the manufacturer’s instructions. Mitochondrial protein concentrations were measured using the Pierce™ Protein Assay Kit (Thermo Scientific, Waltham, MA, USA, Cat#23227). To confirm the lack of living cells in the mitochondria suspension, we incubated the suspension in a plastic dish in an incubator 37 °C. Under these conditions, no cardiac cells we found to adhere after several days of culture. MSCs were exposed to increasing concentrations of cardiac mitochondria corresponding to 0.02 mg, 0.08 mg, and 0.2 mg of proteins per 10^5^ cells (indicated, respectively, as Mito1, Mito2, and Mito3 concentrations) for 24 h in FBS-free DMEM.

To assess the transfer of isolated cardiac mitochondria to MSCs, cardiac mitochondria were isolated from RL14 cells previously labeled with MitoTracker Green FM (40 nM, Invitrogen, Waltham, MA USA, Cat#M7514) and then exposed for 24 h to MSCs. To prevent the staining of MSC mitochondria by the MitoTracker Green released by isolated cardiac mitochondria into the medium during the preconditioning step, cardiac cells were washed 3 times with PBS after MitoTracker staining, maintained for 24 h in a growth medium, and washed again 3 times before we proceeded to mitochondria isolation. 

The presence of cardiac mitochondria in MSCs was then determined on the basis of the MitoTracker Green fluorescence, detected with a LSR Fortessa X20 flow cytometer or LSM800 confocal microscopy (Zeiss, Oberkochen, Germany). Just before confocal microscopy analysis, MSCs were incubated with wheat germ agglutinin (WGA) conjugated to Alexa Fluor 647 (5 μg/mL; Invitrogen, Waltham, MA, USA, Cat#W32466) after their exposure to isolated MitoTracker-stained mitochondria to facilitate their detection.

### 2.3. Inhibition of Cardiac Mitochondria Transfer to MSCs

To characterize the endocytosis process by which MSCs internalize cardiac mitochondria, human MSCs were exposed to cardiac mitochondria previously labeled with MitoTracker Green FM (40 nM, Invitrogen, Waltham, MA, USA, Cat#M7514) in the presence of the dynamin-dependent, clathrin-mediated endocytosis inhibitor dynasore (50 mM, Santa Cruz Biotechnology, Dallas, TX, USA, Cat#sc-202592). After 24 h of treatment, the MitoTracker Green FM (40 nM, Invitrogen, Waltham, MA, USA, Cat#M7514) fluorescence of MSCs was analyzed with flow cytometry or LSM800 confocal microscopy (Zeiss, Oberkochen, Germany).

### 2.4. Inhibition of MSC Autophagy

To inhibit autophagy, MSCs were treated for 3 h with chloroquine (100 μM, Sigma-Aldrich, Saint Louis, MO, USA, Cat#C6628) in serum-free DMEM prior to treatment with cardiac mitochondria. 

### 2.5. Cardiac Mitochondria-Derived ROS Analysis

To determine the ROS levels released by cardiac mitochondria following their transfer to MSCs, cardiac mitochondria were isolated from RL14 cells previously stained with MitoSOX (5 μM, Invitrogen, Waltham, MA, USA, Cat#M36008) for 10 min. MSCs were then exposed to MitoSOX-labeled cardiac mitochondria for 24 h. MitoSOX fluorescence measurements were carried out with a Tecan Infinite M200 Pro plate reader combined with the acquisition software Magellan™ 7.2.

To scavenge ROS, cardiac mitochondria were treated with mitoTEMPO (500 µM, Sigma-Aldrich, Saint Louis, MO, USA, Cat#SML0737) in serum-free DMEM for 1 h prior to their exposure to MSCs.

### 2.6. Cardiac Mitochondria Mitophagy

To assess the degradation of cardiac mitochondria following their internalization in MSCs, cardiac mitochondria were isolated from RL14 cells previously stained with Mtphagy dye for 30 min (100 nmol/L, Dojindo, Rockville, MD, USA, Cat#MD01-10) prior to their exposure to MSCs. After 24 h, Mtphagy fluorescence was measured with a Tecan Infinite M200 Pro plate reader combined with the acquisition software Magellan™ 7.2.

### 2.7. Transmission Electron Microscopy

MSCs were seeded at 5 10^3^ cells/cm^2^ on glass slides coated with 0.2% gelatin (Millipore, Burlington, MA, USA, Cat#104078). After 24 h of exposure to cardiac mitochondria, MSCs were fixed with 3% glutaraldehyde (Sigma-Aldrich, Waltham, MA, USA, Cat#49629) in a 0.1 M sodium phosphate buffer (pH 7.4), post-fixed for 1 h with 1% osmium tetroxide (Electron Microscopy Science, Hatfield, PA, USA, Cat#19190), dehydrated with successive ethanol washes (70%, 90%, 100%, 100%) (Carlo Erba, Val-de-Reuil, France, Cat#528131), and impregnated with epoxy resin (Electron Microscopy Science, Hatfield, PA, USA, Cat#14120). After polymerization, 80–90 nm sections were cut with a Reichert Ultracut S ultramicrotome, stained with 2% uranyl acetate plus Reynold’s lead citrate (Leica Biosystems, Wetzlar, Germany), and visualized under a JEOL 1011 transmission electron microscope with GATAN Erlangshen CCD camera.

### 2.8. Real-Time PCR Assays

RNAs were extracted from cultured MSCs or mouse hearts using the NucleoSpin^®^ RNA spin kit (MACHEREY-NAGEL GmbH & Co. KG, Düren, Germany, Cat#740955.250) or the TRIzol™ Reagent (Invitrogen, Waltham, MA, USA, Cat#15596026), respectively. RNAs were then reverse-transcribed with SuperScript™ III Reverse Transcriptase (Invitrogen, Waltham, MA, USA, Cat#18080044) and random primers. Quantitative RT-PCR (qPCR) reactions were performed in duplicate using the PowerUp™ SYBR™ Green Master Mix (Applied Biosystems, Waltham, MA, USA, Cat#A25742) and the StepOnePlus™ detection system (Applied Biosystems, Waltham, MA, USA) associated with the acquisition and analysis StepOne™ software v2.1 (Applied Biosystem, Waltham, MA, USA). The PCR conditions were as follows: 10 min at 95 °C, followed by 40 cycles comprising one step at 95 °C for 15 s and one step at 60 °C for 1 min, and a final cycle of 15 s at 95 °C, 1 min at 60 °C and 15 s at 95 °C. The human and mouse TATA-Binding Protein (TBP) mRNAs were used as reference genes for the quantification of human and mouse transcripts of interest, respectively. The sequences of the used primers are listed in Appendix A.

### 2.9. Collection of MSC Conditioned Media

MSCs were seeded at 5 × 10^3^ cells/cm^2^ prior to exposure to cardiac mitochondria in the FBS-free medium. After 24 h of exposure, the supernatants were collected, centrifuged at 3000 g for 15 min to remove cell debris and mitochondria, and kept frozen until use.

### 2.10. ELISAs 

The secretion levels of VEGF-A and HGF by MSCs following cardiac mitochondria transfer were measured with enzyme-linked immunosorbent assays (ELISAs) (PeproTech, Cranbury, NJ, USA, Cat#BGK14210, Cat#900-K10, Cat#900-K00) according to the manufacturer’s instructions. Absorbance measurements were carried out with a Tecan Infinite M200 Pro plate reader combined with the acquisition software Magellan™ 7.2. Cytokine concentrations were calculated from the four-parameter logistic curve obtained from serial dilutions of standard proteins.

### 2.11. Luminex Assays

The secretion levels of a panel of MSC pro-angiogenic factors, chemokines, and cytokines following cardiac mitochondria transfer were quantified with Human Angiogenesis/Growth factor Magnetic Bead Panel 1 Milliplex MAP kit (Merck-Millipore, Burlington, MA, USA) or Human Luminex Discovery Assay kit (Bio-Techne, Minneapolis, MN, USA) according to the manufacturers’ instructions. The median fluorescence intensity for each sample was measured in duplicate using the Bio-Plex 200 system (Bio-Rad, Hercules, CA, USA). The Bio-Plex Manager software version 6.1 (Bio-Rad, Hercules, CA, USA), incorporating a weighted five-parameter logistic curve-fitting method, was used to calculate the cytokine concentration.

### 2.12. Collagenase Enzymatic Activity

The collagenase activity of MSCs untreated or treated with various concentrations (0.02 mg, 0.08 mg, or 0.2 mg of protein/1.5 × 10^5^ cells) of human-isolated mitochondria was evaluated on cell lysates using the Collagen Degradation/Zymography Assay Kit (Abcam, Cambridge, UK, Cat#ab234624) according to the manufacturer’s instructions. Absorbance measurements were carried out with a Tecan Infinite M200 Pro plate reader combined with the acquisition software Magellan™ 7.2.

### 2.13. Murine Model of Myocardial Infarction and Cell Grafting

We randomized 8–10-week-old C57BL/6J male mice into 3 groups (*n* = 11–16): infarcted mice engrafted with MSCs preconditioned with cardiac mitochondria (0.5 mg) for 24 h, infarcted mice engrafted with untreated MSCs, and infarcted mice injected with the vehicle (HBSS) alone. Left anterior descending (LAD) coronary artery ligations were induced under general anesthesia with an isoflurane vaporizer and oxygen supply. Mice were placed in an induction chamber with a 1 L/min O_2_ flow rate and 3% isoflurane flow over approximately 4 min. The maintenance of anesthesia was performed with 2% isoflurane inhalation with analgesia (buprenorphine 0.1 mg/kg°s.c.) and ventilated by orotracheal intubation (Minivent, Harvard Apparatus) with a controlled stroke (150 μL) and frequency (140/min).

After left thoracotomy and muscular dissection, the ligation of the left coronary artery was performed with 8–0 silk to obtain a permanent ischemia. The ischemia was confirmed by the sudden regional paleness of the myocardium and ST elevation. Either 7 × 10^5^ MSCs preconditioned with cardiac mitochondria (0.2 mg of protein/1.5 × 10^5^ cells) for 24 h or 7 × 10^5^ untreated MSCs or vehicle (HBSS, Thermo Scientific, Waltham, MA, USA Cat#14025050) were injected using a Hamilton syringe in four regions of the left ventricle at the border zones of the infarcted area for a total volume of 20 μL. Muscle and cutaneous plans were sutured with 6–0 silk. Mice were extubated and placed at 32 °C (RT) for 1 h. This study was approved by the local ethics committee for animal experimentation and registered by the national committee under the number CEEA-LR-28731. Mice were euthanized at day 3, day 7, and day 21 for heart removal.

### 2.14. Echocardiography and Speckle Tracking Analysis

High-resolution echocardiography (Vevo-3100, VisualSonics/Fujifilm, Toronto, Canada, with a 40 MHz MX550D ultrasound probe) was used to assess left ventricle (LV) function (ejection fraction and longitudinal strain) at 1 day, 7 days and 21 days after surgery according to the European Society of Cardiology Working Group on Myocardial Function [26]. Briefly, cardiac function was assessed under general anesthesia by 2% isoflurane inhalation. The ECG and respiratory rate were monitored during the procedure. Multiple four-dimensional B-mode parasternal short-axis views from apex to base were recorded to measure the LV ejection fraction (LVEF). After tracing endocardial end-diastolic and end-systolic areas, 3D LV volumes were calculated with Simpson’s method of disks, and LVEF was determined with the following formula: (end-diastolic LV volume – end-systolic LV volume)/(end-diastolic LV volume). Global and regional longitudinal strain analyses were performed under the 2D parasternal long-axis view. Offline image analysis was performed using dedicated VisualSonics Vevolab 5.6.1 and Vevostrain software v5.7.1

### 2.15. Photoacoustic Imaging

The Vevo LAZR-X (VisualSonics/Fujifilm, Toronto, Canada) system was used in this study to generate photoacoustic (PA) images to generate the real-time vascular oxygen saturation (sO_2_) mapping of the LV anterior wall using the PA EKV oxy-hemo mode [27]. For the PA signal acquisition, the gain was set to 34 dB with a 2D gain of 22 dB. A B-mode analysis was also performed for the colocalization of the photoacoustic signals. The calculation of sO_2_ was achieved offline with the VevoLAB software by selecting areas of interest (myocardial anterior wall) in the B mode.

### 2.16. Statistical Analysis

Data analysis was performed using GraphPad Prism software version 9.3.1 (San Diego, CA, USA). For the statistical analysis of data obtained from *n* ≥ 5 independent experiments, parametric tests were used after the validation of normality and equality of variances of populations using the Shapiro–Wilk and Browne–Forsythe tests, respectively. Otherwise, non-parametric tests including the Mann–Whitney test and a one-way ANOVA on rank followed by Dunn’s multiple comparisons were used. For the statistical analysis of data obtained from *n* < 5 experiments, non-parametric tests were applied. The statistical method used for each experiment is indicated in each figure legend.

For in vivo studies, mice were randomly assigned to the different treatment groups. Investigators were blinded during the analysis. No statistical method was used to predetermine sample size.

## 3. Results

### 3.1. Transfer of Cardiac Mitochondria Stimulates Proliferation and the Regenerative Properties of Recipient MSCs

We previously reported that MSCs are able to capture free mitochondria isolated from cardiomyocytes or platelets [16,17]. To confirm the occurrence of this phenomenon in our experimental settings, mitochondria from the cardiac RL14 cell line previously labeled with MitoTracker were isolated and incubated at different concentrations with MSCs. After 24 h of exposure, and consistent with our earlier observations, confocal microscopy revealed that in their cytoplasm, MSCs contained cardiac mitochondria with a typical network distribution around the nucleus (Figure 1A). The presence of cardiac mitochondria was also detected in MSCs with flow cytometry (Figure 1B). The quantification of MitoTracker fluorescence through this method indicated that MSCs efficiently engulfed cardiac mitochondria in a dose-dependent manner (Figure 1B). 

We then investigated whether cardiac mitochondria transfer had an impact on MSCs’ functions that are critical for their therapeutic use. 

We found that the cardiac mitochondria preconditioning of MSCs increased the mRNA levels of the proliferation marker Ki67 and the pro-angiogenic factors VEGF and HGF compared with untreated MSCs (Figure 1C,D). We also observed that cardiac mitochondria transfer enhanced the MSCs’ secretion of the pro-angiogenic proteins endoglin, FGF-1, HGF, IL8, PLGF, VEGF-A, VEGF-C, and VEGF-D (Figure 1E); the chemoattractant chemokines CXCL1, CXCL5 and CXCL6; and the anti-inflammatory cytokines IL11, IL33 and LIF (Figure 1F). Finally, we showed that the mitochondria-preconditioning of MSCs upregulated their transcriptional expression of the metalloproteinases MMP1, MMP9 and MMP14 (Figure 1G); their secretion of MMP1 (Figure 1H); and their collagenase activity (Figure 1I), indicative of an overall increased capacity of preconditioned MSCs to degrade the extra-cellular matrix. Taken together, these findings indicate that the transfer of cardiac mitochondria to MSCs enhances their proliferative, pro-angiogenic, immuno-modulatory and anti-fibrotic properties.

### 3.2. Free Cardiac Mitochondria Are Transferred to MSCs via Dynamin-Dependent, Clathrin-Mediated Endocytosis

We then investigated the process by which MSCs internalize free cardiac mitochondria. We previously reported that platelet-derived mitochondria are internalized by MSCs through dynamin-dependent, clathrin-mediated endocytosis [17]. We therefore examined whether a similar process was involved in the capture of exogenous cardiac mitochondria by MSCs. MSCs’ exposure to cardiac mitochondria, previously labeled with MitoTracker Green, was performed in the presence of dynasore, an inhibitor of dynamin-dependent, clathrin-mediated endocytosis. Confocal microscopy and flow cytometry analyses revealed that dynasore treatment almost completely suppressed the transfer of cardiac mitochondria to MSCs (Figure 2A,B). Following dynasore treatment, cardiac mitochondria incubation with MSCs also failed to increase their transcriptional expression of the proliferative marker Ki67 (Figure 2C); the pro-angiogenic factors VEGF and HGF (Figure 2D); the chemoattractant chemokines CXCL1, CXCL5 and CXCL6; the anti-inflammatory cytokines IL11, IL33, and LIF (Figure 2E); and the metalloproteinases MMP1, MMP9 and MMP14 (Figure 2F). In the presence of dynasore, the increased collagenase activity of mitochondria-pretreated MSCs was also inhibited (Figure 2G). Overall, these results indicated that cardiac mitochondria were internalized by MSCs through dynamin-dependent, clathrin-mediated endocytosis. They also reinforced the role of exogenous cardiac mitochondria to potentiate the activation and regenerative capacities of MSCs.

### 3.3. Degradation of Cardiac Mitochondria Transferred to MSCs Is Necessary to Trigger Their Therapeutic Potential

To investigate the fate of cardiac mitochondria after their internalization by MSCs, we first performed transmission electron microscopy on MSCs treated or not with cardiac mitochondria. MSCs preconditioned with cardiac mitochondria for 24 h contained a greater number of mitochondria trapped in phagosome-like vesicles than untreated MSCs (Figure 3A), suggesting that cardiac mitochondria were degraded in MSCs by mitophagy. To test this hypothesis, we labeled cardiac mitochondria prior to their exposure to MSCs with Mtphagy dye, a dye emitting fluorescence following the fusion of mitochondria with lysosomes. After 24 h of exposure, Mtphagy fluorescence was detected in the recipient MSCs, indicative of the degradation of cardiac mitochondria (Figure 3B). To assess whether this process was required for the functional activation of recipient MSCs, we performed the mitochondria preconditioning of MSCs pretreated with chloroquine, an autophagy inhibitor. In these conditions, cardiac mitochondria failed to stimulate the proliferation of recipient MSCs, as demonstrated by the lack of increased Ki67 mRNA expression in chloroquine-pretreated MSCs following cardiac mitochondria transfer compared with untreated counterparts (Figure 3C). Similarly, in chloroquine-pretreated MSCs, cardiac mitochondria exposure failed to increase the mRNA levels of pro-angiogenic factors (Figure 3D), chemokines (Figure 3E, upper panels), anti-inflammatory cytokines (Figure 3E, lower panels), and metalloproteinases (Figure 3F). At the protein level, cardiac mitochondria treatment was also less effective in stimulating the secretion of the pro-angiogenic factor HGF (Figure 3G), chemokines (CXCL1, CXCL5 and CXCL6) and anti-inflammatory cytokines (IL11 and LIF) (Figure 3H) when MSCs were pretreated with chloroquine. Finally, the chloroquine-pretreatment of MSCs also abolished the increase in collagenase activity observed following the cardiac mitochondria preconditioning of MSCs (Figure 3I). 

Overall, these results highlight the critical role of mitophagy in promoting MSC activation following mitochondria preconditioning.

### 3.4. ROS Produced by Cardiac Mitochondria Trigger Autophagy and Activation of Recipient MSCs

Since mitochondrial ROS have been reported as activators of mitophagy [28], we examined whether the ROS produced by cardiac mitochondria could trigger their own degradation in recipient MSCs and lead to their activation. With this purpose, cardiac mitochondria were stained with MitoSOX red dye prior to their exposure to MSCs [29]. Following the internalization of cardiac mitochondria, the detected MitoSOX fluorescence indicated the generation ROS species in the recipient MSCs (Figure 4A). To determine whether the generation of these ROS was responsible for the increased autophagy observed in recipient MSCs, in addition to Mtphagy dye, cardiac mitochondria were also pre-incubated with mitoTEMPO (a ROS scavenger) prior to their exposure to MSCs. The pretreatment of cardiac mitochondria with mitoTEMPO inhibited the increased Mtphagy-dependent fluorescence in mitochondria-preconditioned MSCs (Figure 4B). This indicated that cardiac mitochondria-induced ROS production supports ROS degradation in MSCs. The changes in the Ki67 mRNA levels were not statistically significant in these conditions (Figure 4C). However, following mitoTEMPO pre-treatment, cardiac mitochondria failed to increase the transcription and secretion levels of the pro-angiogenic factors VEGF-A and HGF and the chemokines CXCL1, CXCL5 and CXCL6 (Figure 4D,G). The transcriptional expression levels of the anti-inflammatory cytokines IL11, IL33 and LIF, as well as the secretion levels of IL11, were also significantly reduced in reference to MSCs preconditioned with untreated cardiac mitochondria (Figure 4F,G). Likewise, mitoTEMPO-treated cardiac mitochondria, in comparison to untreated mitochondria, were less efficient in stimulating the transcriptional expression of the metalloproteinases MMP1, MMP9 and MMP14 in MSCs (Figure 4H). On the other hand, measurements of the collagenase activity of MSCs following conditioning with cardiac mitochondria, pre-treated or not with mitoTEMPO, did not show any differences (Figure 4I). Overall, the ROS produced by cardiac mitochondria contribute to their own degradation in recipient MSCs, leading to their activation, a secretory shift towards a regenerative phenotype, and the activation of a number of factors associated with MSCs’ regenerative properties.

### 3.5. Transfer of Cardiac Mitochondria Enhances the Therapeutic Potential of Transplanted MSCs in a Mouse Model of Ischemic Cardiomyopathy

On the basis of the enhanced MSC regenerative profile observed in vitro, we next asked whether the preconditioning of MSCs with cardiac mitochondria could improve their cardioprotective functions when grafted in vivo. With this objective, we injected either naive or mitochondria-preconditioned MSCs into the left ventricular (LV) border zone of infarcted mouse hearts immediately after LV coronary ligation. First, we examined the survival rate and activation of engrafted human MSCs using RT-qPCR with human-specific primers. At day 3 post-surgery, the mRNA levels of human TATA-Binding Protein (TBP) (Figure 5A) and Ki67 (Figure 5B) were significantly higher in LVs transplanted with mitochondria-preconditioned MSCs compared with naive MSCs, indicating that engrafted mitochondria-preconditioned MSCs maintained higher survival and proliferative rates than their naive counterparts. In addition, the levels of human mRNAs encoding the pro-angiogenic factors VEGF-A and IL8 (Figure 5C); CXCL1, CXCL5, and CXCL6 chemokines (Figure 5D); anti-inflammatory cytokine LIF (Figure 5D); and metalloproteinase MMP14 (Figure 5E) were also higher for mitochondria-preconditioned MSCs compared with naive MSCs. These data supported the fact that the mitochondria-preconditioned MSCs had enhanced angiogenic and anti-inflammatory potential compared with their naive counterparts following their engraftment in mouse myocardial infarcts. 

Next, we examined how the murine heart tissues responded to infarct injury, following human MSC injection into the injured heart, by measuring the expression of genes of murine origin related to tissue repair at day 3, day 7, and day 21 post-injection. At all time-points after MI, infarcted hearts injected with MSCs preconditioned with cardiac mitochondria (Mito 3) expressed significantly higher mouse mRNA levels of the pro-angiogenic factors VEGF, VEcadherin (except for day 3 post-MI) and CD31 than mock-treated (HBSS) hearts (Figure 6A). Importantly, as observed at day 7 and day 21, cardiac mitochondria-preconditioned MSCs were more effective than naive MSCs (NT) at enhancing the transcription rates of VEGF, VEcadherin and CD31 (Figure 6A). Likewise, at day 7 post-MI, the expression levels of mRNAs encoding the anti-inflammatory LIF cytokine and the pro-healing M2 macrophage markers Arginase 1 (Arg1), CD206, CHIL3 and CHIA were higher in the LVs injected with cardiac mitochondria-preconditioned MSCs than with naive MSCs or a saline HBSS (Figure 6B). The reduction in the transcriptional expression levels of the inflammation marker iNOS and the inflammatory cytokine TNFα were also more effective with mitochondria-preconditioned MSCs than with naive MSCs or HBSS, as observed at day 7 and day 21 post-MI, with an effect observed as early as day 3 for iNOS (Figure 7C). Finally, at day 21 post-MI, the murine mRNA expression of the three extracellular matrix components collagen-1 (Col-1), collagen-3 (Col-3), and fibronectin-1 was also diminished in LVs injected with mitochondria-preconditioned MSCs compared with those injected with non-preconditioned MSCs (Figure 6D). Taken together, these results indicated that the mitochondria-preconditioning of MSCs improves their therapeutic potential following engraftment in infarcted hearts by fostering angiogenesis and counteracting inflammation and fibrosis in ischemic hearts.

### 3.6. Mitochondria-Preconditioned MSCs Improve Post-MI LV Function and Perfusion

We tested whether mitochondria-preconditioned MSCs were more efficient than naive MSCs in improving post-MI cardiac function after intracardiac injection. We found that, like naive MSCs, MSCs preconditioned with cardiac mitochondria improved global LV function in ischemic hearts at day 21 post-MI, as evidenced by an increased LV ejection fraction compared with mock-treated hearts (Figure 7A). 

When measuring the heart global longitudinal strain, mitochondria-preconditioned MSCs appeared to be even more effective in decreasing this strain than naive MSCs (Figure 7B). The same mitochondria-enhanced effect was observed for segmental strain analysis, as cardiac mitochondria-preconditioned MSCs decreased the apical and mid anterior LV wall longitudinal strain of infarcted hearts more effectively than naive MSCs (Figure 7C). Finally, infarcted hearts injected with naive or cardiac mitochondria-preconditioned MSCs exhibited similar decreases in basal anterior LV longitudinal strain compared with mock-treated hearts (Figure 7D). In order to evaluate the impact of MSC injection on LV perfusion, we used myocardial photoacoustic imaging [27,30] to detect oxygen saturation (sO_2_) changes in the infarct area (apical and mid LV wall) at day 21 post-MI. It is to be noted that only the cardiac mitochondria-preconditioned MSCs increased sO_2_ values in the LV ischemic wall (Figure 7E,F). Taken together, these data demonstrate that the cardiac mitochondria-preconditioning of MSCs enhances the therapeutic benefit of their delivery in the ischemic heart by improving LV function perfusion, as observed at day 21 post-injection.

## 4. Discussion

The discovery that mitochondria can traffic between cells, altering the behavior of recipient cells and, for example, promoting tissue regeneration, has fundamentally changed our perception of the diverse functions of these organelles. Our study shows how intercellular mitochondria transfer can be advantageously exploited for the design of novel therapeutic approaches. Indeed, our data provide compelling evidence that the therapeutic efficacy of MSCs for ischemic heart treatment can be optimized via the prior transfer of cardiac mitochondria. Of paramount interest, we showed that artificial cardiac mitochondrial transfer to MSCs deeply alters their reparative activity. Among these alterations, the proliferation and pro-angiogenic, anti-inflammatory and anti-fibrotic properties of MSCs were enhanced in direct connection with their enhanced capacity to repair mouse myocardial infarcts. We previously reported that mitochondria transfer stimulates the cytoprotective and pro-angiogenic functions of MSCs [16,17]. Consistent with these findings, our present study further reveals that mitochondrial transfer has a much broader impact on MSC functionality than previously documented by highlighting its role in regulating the anti-inflammatory and anti-fibrotic properties of MSCs. 

Our research also provides new insights into how cardiac mitochondria transfer activates the properties of MSCs. In particular, our study indicates that the activation of MSCs is driven by the ROS-dependent mitophagy of exogenous mitochondria. Interestingly, ROS-dependent mitophagy was also observed in MSCs following their coculture with damaged cardiac or endothelial cells and the transfer of the mitochondria from these cells to MSCs [16]. 

Our results pave the way for future prospects to identify the mitophagy-triggered signaling pathways that mediate the pro-stimulatory effects of mitochondria transfer on MSCs’ repair functions. In particular, one of the key questions to be answered is whether the mitophagy of transferred mitochondria alters the metabolism of recipient MSCs and whether these metabolic changes are involved in enhancing their repair properties.

The use of mitochondria as therapeutic tools has already attracted considerable interest for the treatment of several diseases, including ischemic myocardial injury, stroke, and lung and liver injury [31]. This therapeutic mitochondria-based regenerative approach has emerged from a series of studies reporting that MSCs were able to improve the bioenergetics and the survival of damaged cells through the transfer of their mitochondria [12,32,33]. Based on these observations, the transplantation of pre-isolated mitochondria has been envisioned as a strategy to mitigate tissue injury. In particular, the transplantation of exogenous mitochondria has been reported to improve cardiac function in mouse, rat, rabbit and pig models of MI, with ongoing clinical trials [31,34,35,36,37,38]. Even if the mechanisms whereby transplanted mitochondria improve heart function remain to be formally elucidated, exogenous mitochondria have been shown to be internalized by ischemic cardiomyocytes, leading to the preservation of cardiomyocyte energy metabolism, viability and function [36]. However, despite its promising potential, the clinical application of this technique to a wide range of patients is seriously hampered by several critical issues. Most notably, the preservation of the integrity and function of the pre-isolated mitochondria in the extracellular environment and the controlled efficiency of their internalization into cardiomyocytes remain questionable. 

Although the status of isolated mitochondria was not examined in the present study, the method we used has been described by several publications as enabling the isolation of intact and functional mitochondria from cultured cells or tissues [39] Available online 10 December 2019, Version of Record 14 March 2020). Whether isolated mitochondria need to be functional to activate MSCs remains an open question. However, according to our previous data showing that mitochondria transferred by platelets or damaged cardiomyocytes/endothelial cells to MSCs have to contain mtDNA or be respiratory competent to activate MSCs [16,17] it seems that exogenous mitochondria have to be intact to trigger ROS boost in MSCs and, therefore, to stimulate their repair properties. 

Mitochondria trafficking from MSCs to surrounding damaged cells has also been shown to have pro-healing effects, so methods to enhance this trafficking, notably through Miro-1 overexpression, have been considered promising approaches to optimize the efficacy of MSC-based therapy [32,40]. In our study, as a first evaluation, we linked the therapeutic efficacy of cardiac mitochondria-conditioned MSCs to their increased secretion of a number of pro-angiogenic, chemoattractant and anti-inflammatory factors. However, because many of the MSC properties were altered under these conditions, it is also conceivable that the capacity of MSCs to transfer their mitochondria to surrounding injured cardiac cells was also increased. This process could contribute, in combination with paracrine factor secretion, to their enhanced therapeutic efficacy. This issue was not addressed in the present study. However, we previously reported that mitochondria transferred from damaged cardiomyocytes to MSCs under coculture settings, which recapitulate the effects of cardiac mitochondria preconditioning, stimulated the trafficking of mitochondria in the opposite direction, i.e., from MSCs to damaged cells [16]. Interestingly, in another study, we also observed that following their engraftment into mouse ischemic hearts, human MSCs transferred their mitochondria to cardiomyocytes in vivo and that MSCs (having received mitochondria following coculture with distressed mouse cardiomyocytes) had the same capacity [10]. This suggests that mitochondria transfer from MSCs to injured cardiomyocytes could also occur and be enhanced following therapy with cardiac mitochondria-preconditioned MSCs in damaged hearts, thus contributing to tissue repair. Determining the respective contributions for heart repair of mitochondria trafficking and cytokine secretion from cardiac mitochondria-conditioned MSCs remains to be determined.

The therapeutically targeted process of the mitochondrial preconditioning of MSCs, which we presented in this study, displays notable advantages in terms of safety, feasibility and efficacy. Still, a number of questions remain to be addressed for clinical considerations and applications. A first question concerns the cellular source for mitochondrial isolation with the goal of reaching maximal activation of MSCs’ repair capacities. On the basis of our previous findings, we expect this parameter to be of primary importance for the efficacy of the mitochondria preconditioning procedure. We found that, depending on their cellular origin, transferred mitochondria differently altered MSC properties. For instance, platelet-derived mitochondria were found to improve the angiogenic potential of recipient MSCs but not to affect MSC proliferation and immunomodulatory properties, in contrast to what we demonstrated here for cardiac mitochondria [17]. Another issue is related to the therapeutic window and engraftment strategy, which showed therapeutic benefit in our experimental protocol but will presumably need to be optimized. In our preclinical study, MSCs were delivered via direct point injections in the border zone at the onset of the myocardial infarction. However, defining the most effective route for MSC delivery to maximize their therapeutic effects remains a challenging question. Among the different strategies, transendocardial injection showed superior efficacy on infarct size and functional improvement compared with direct intramyocardial injection in acute MI [41]. As the delivery procedure will directly impact the myocardial retention of MSCs, other strategies including the use of biomaterials or scaffolds may also be helpful [42,43]. To date, no study has formally compared the different possible routes for MSC administration regardless of the type of ischemia, acute or chronic. Optimizing the delivery of MSCs is certainly one of the most challenging obstacles to overcome to design new clinical studies in the near future [44,45].

Finally, one of the most critical issues raised by our study and related to clinical applications concerns the high functional variability of MSCs according to the age and health of donors, thus requiring the rigorous quality control of MSCs. To circumvent this problem, mitochondria preconditioning should be performed for MSCs derived from pluripotent stem cells (iPSC-MSCs) that have strong immunomodulatory properties, can be produced at the clinical grade [46] doi: 10.1007/978-1-4939-3584-0_17, and have already been tested in clinical trials [47].

To conclude, the intercellular exchange of mitochondria provides a stimulating and promising field of investigation to improve the therapeutic potential of MSCs. Here, we have proposed a novel paradigm whereby the preconditioning of MSCs with mitochondria has been shown to boost their therapeutic efficacy. Obviously, there are still unanswered questions concerning, for instance, the precise mechanisms responsible for the activation of mitophagy or its potential impact on MSC metabolism and functionality. Still, our study provides an important advance for the optimization of future cell-based therapies in ischemic diseases.

## Figures and Tables

**Figure 1 cells-12-00582-f001:**
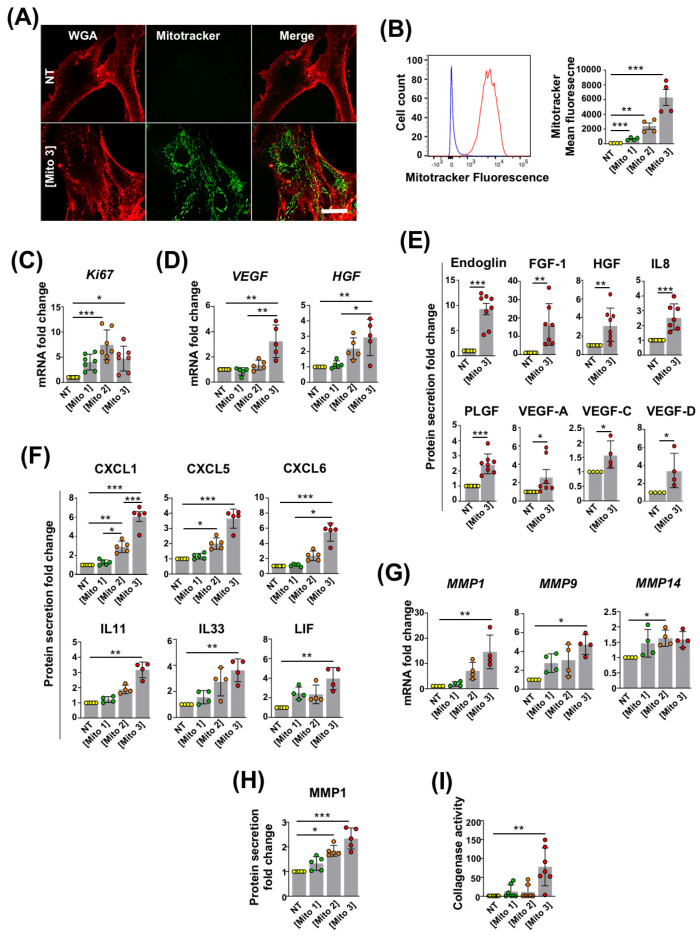
Cardiac mitochondria transfer alters MSC properties. (**A**) Representative confocal microscopy images of WGA-stained MSCs in the absence (NT) or presence of MitoTracker Green-labeled cardiac mitochondria at the Mito 3 concentration after 24 h of incubation. Scale bar: 5 µm. (**B**) Internalization of MitoTracker Green-labeled cardiac mitochondria by MSCs following 24 h of exposure to different cardiac mitochondria concentrations (Mito1: 0.02 mg/1.5 × 10^5^ cells; Mito2: 0.08 mg/1.5 × 10^5^ cells; Mito3: 0.2 mg/1.5 × 10^5^ cells). Left panel shows a representative flow cytometry histogram (blue: untreated MSCs; red: MSCs treated with the Mito 3 mitochondria concentration). Right panel shows flow cytometry quantification (*n* = 4). (**C**) Relative *Ki6*7 mRNA levels in cardiac mitochondria-preconditioned MSCs in reference to non-treated MSCs (*n* = 7). (**D**) Relative *VEGF* and *HGF* mRNA levels in cardiac mitochondria-preconditioned MSCs in reference to non-treated MSCs (*n* = 5). (**E**) Relative protein secretion levels of endoglin, FGF-1, HGF, IL8, PLGF, VEGF-A (*n* = 8) or VEGF-C and VEGF-D (*n* = 4) in reference to non-treated MSCs. (**F**) Relative protein secretion levels of CXCL1, CXCL5, CXCL6, IL11, IL33 and LIF in conditioned media from cardiac mitochondria-preconditioned MSCs (Mito 3) relative to non-treated MSCs (*n* ≥ 4). (**G**) Relative mRNA levels of *MMP1*, *MMP9* and *MMP*14 (*n* = 4); (**H**) MMP1 secretion levels (*n* = 5); and (**I**) collagenase activity (*n* = 7) in cardiac mitochondria-preconditioned MSCs in reference to non-treated MSCs. One-way ANOVA with Dunn’s multiple comparison test in (**B**–**D**, **F**–**I**). Unpaired t-test in (**E**). * *p <* 0.05, ** *p* 0.01, *** *p <* 0.001. Each dot represents an independent experiment. Bar graphs represent mean values ± SD.

**Figure 2 cells-12-00582-f002:**
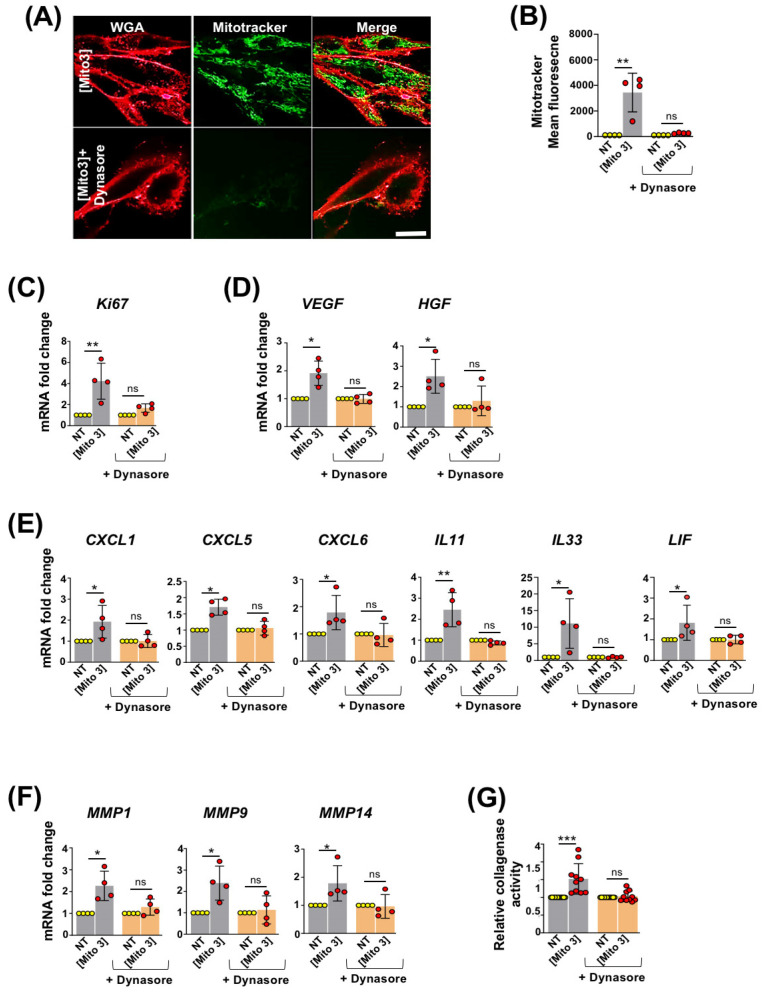
Cardiac mitochondria are internalized by MSCs through dynamin-dependent, clathrin-mediated endocytosis. (**A**) Representative confocal microscopy pictures of WGA-stained MSCs after 24 h of incubation with MitoTracker Green-labeled cardiac mitochondria at the Mito 3 concentration in the absence or presence of dynasore. Scale bar: 5 µm. (**B**) Flow cytometry quantification of MitoTracker Green-labeled cardiac mitochondria by MSCs following 24 h of exposure in the presence or absence of dynasore (*n* = 4). (**C**) Relative *Ki67* mRNA levels in cardiac mitochondria-preconditioned MSCs in the presence or absence of dynasore in reference to their respective controls (*n* = 4). (**D**) Relative *VEGF* and *HGF* mRNA levels in cardiac mitochondria-preconditioned MSCs in the presence or absence of dynasore in reference to their respective controls (*n* = 4). (**E**) Relative mRNA levels of *CXCL1*, *CXCL5*, *CXCL6*, *IL11*, *IL33* and *LIF* in cardiac mitochondria-preconditioned MSCs in the presence or absence of dynasore in reference to their respective controls (*n* = 4). (**F**) Relative mRNA levels of *MMP1*, *MMP9* and *MMP14* (*n* =4) and (**G**) collagenase activity (*n* = 10) in cardiac mitochondria-preconditioned MSCs in the presence or absence of dynasore in reference to their respective controls. One-way ANOVA with Dunn’s multiple comparisons test in (**B**–**F**). One-way ANOVA with Tukey’s multiple comparisons test in (**G**). * *p <* 0.05, ** *p <* 0.01, *** *p <* 0.001. Each dot represents an independent experiment. Bar graphs represent mean values ± SD.

**Figure 3 cells-12-00582-f003:**
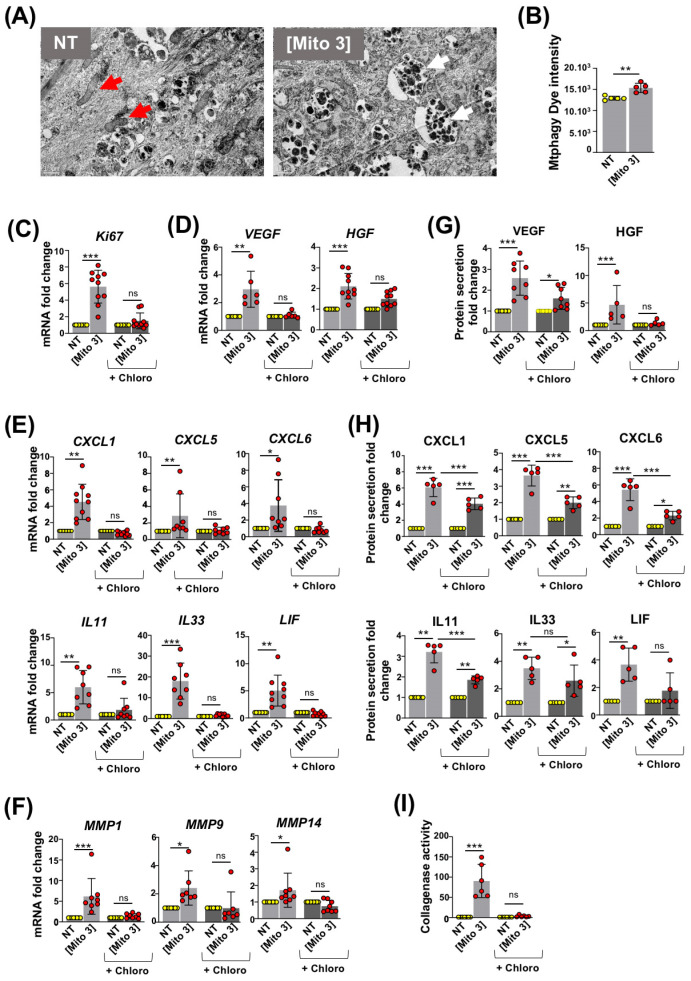
Degradation of cardiac mitochondria is required for MSC activation. (**A**) Transmission electron micrographs taken 24 h after exposure of MSCs to cardiac mitochondria (right panel, Mito 3 concentration; left panel, non-treated MSCs). Red arrows: intact mitochondria. White arrows: autophagolysosomes. Scale bar: 1 µm. (**B**) Mtphagy dye fluorescence intensity in cardiac mitochondria-preconditioned MSCs in reference to untreated ones (*n* = 10). (**C**–**I**) Prior cardiac mitochondria transfer at the Mito 3 concentration, MSCs were treated or not with chloroquine (Chloro) and compared with their respective controls. (**C**–**F**) Relative mRNA levels of (**C**) *Ki67* (*n* = 10); (**D**) *VEGF* (*n* = 6) and *HGF* (*n* =10); (**E**) *CXCL1*, *CXCL5*, *CXCL6*, *IL11*, *IL33* and *LIF* (*n* ≥ 8); (F) *MMP1*, *MMP9* and *MMP14* (*n* ≥ 7). (**G**,**H**) Relative protein secretion levels of (**G**) VEGF (*n* = 8) and HGF (*n* = 5) and (**H**) CXCL1, CXCL5, CXCL6, IL11, IL33 and LIF (*n* = 5). (**I**) Relative collagenase activity (*n* = 6). Unpaired Student’ *t*-test in (**B**). One-way ANOVA with Tukey’s multiple comparisons test in (**C**–**I**). * *p <* 0.05, ** *p <* 0.01, *** *p <* 0.001. Each dot represents an independent experiment. Bar graphs represent mean values ± SD.

**Figure 4 cells-12-00582-f004:**
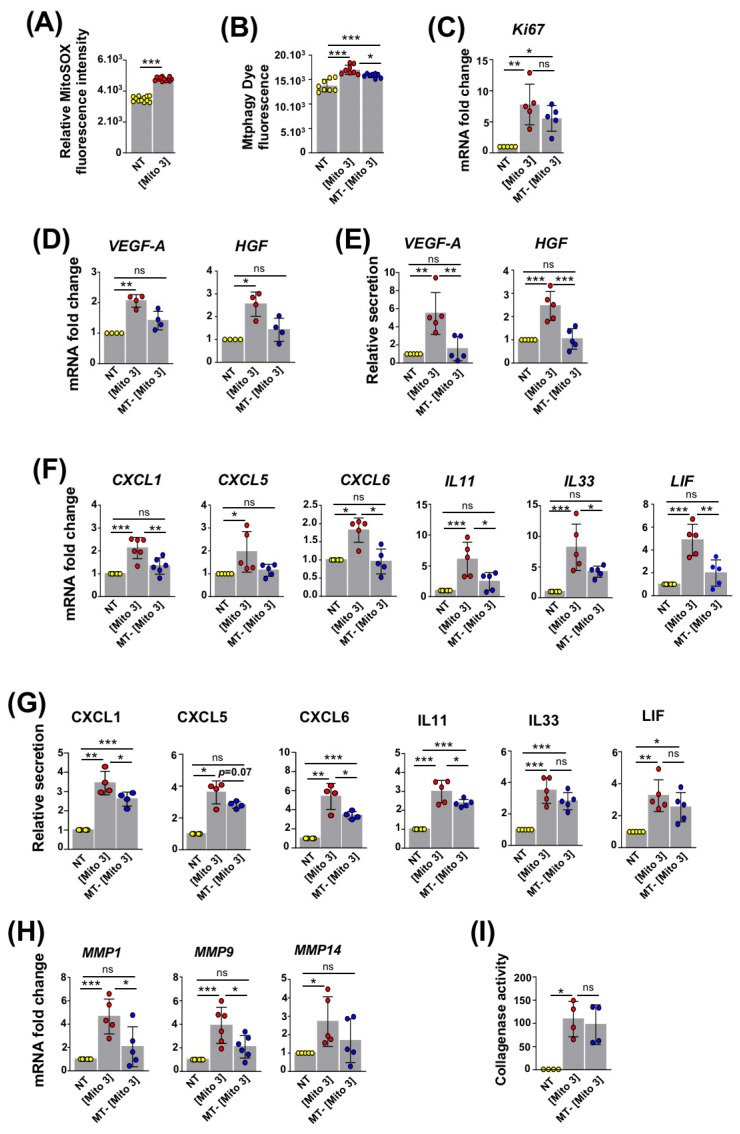
ROS from cardiac mitochondria trigger the mitophagy-dependent activation of MSCs. (**A**) MitoSOX fluorescence in MSCs following exposure to cardiac mitochondria (the Mito 3 concentration) (*n* = 12). (**B**–**I**) MSCs were exposed to cardiac mitochondria (the Mito 3 concentration) previously treated with mitoTEMPO (MT) or not and compared with non-treated MSCs (NT). (**B**) Mtphagy fluorescence (*n* = 8). (**C**) Relative mRNA expression (*n* = 5). (**D**) Relative mRNA (*n* = 4) and (**E**) protein secretion (*n* = 5) levels of VEGF-A and HGF. (**F**) Relative mRNA (*n* ≥ 5) and (**G**) protein secretion (*n* = 4) levels of CXCL1, CXCL5, CXCL6, IL11, IL33 and LIF. (**H**) Relative mRNA expression of *MMP1*, *MMP9* and *MMP14* (*n* ≥ 5). (**I**) Relative collagenase activity (*n* = 4). Unpaired Student’s *t*-test in (**A**). One-way ANOVA with Tukey’s multiple comparisons test in (**B**,**C**,**E**,**F**,**H**). One-way ANOVA with Dunn’s multiple comparisons test in (**D**,**G**,**I**). * *p <* 0.05, ** *p <* 0.01, *** *p <* 0.001. Each dot represents an independent experiment. Bar graphs represent mean values ± SD.

**Figure 5 cells-12-00582-f005:**
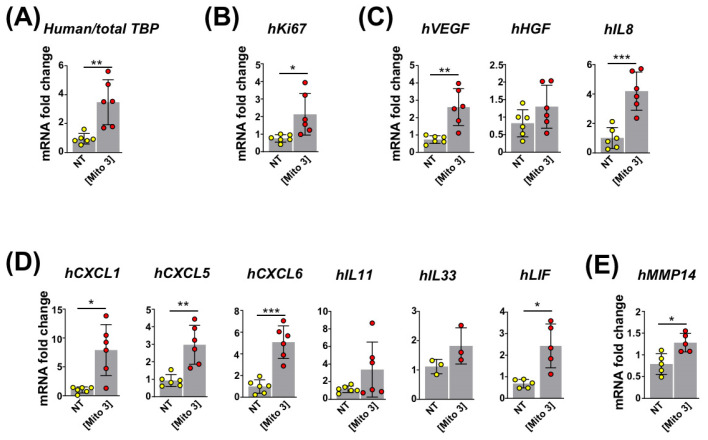
Cardiac mitochondria transfer improves the reparative functions of intramyocardially grafted MSCs. (**A**–**E**) MSCs were conditioned with cardiac mitochondria at the Mito 3 concentration 24 h prior to engraftment in mouse hearts. (**A**) Relative human/total *TBP* mRNA ratios in infarcted mouse myocardia at day 3 post-engraftment of cardiac mitochondria-preconditioned MSCs in reference to hearts injected with non-treated MSCs. (**B**–**E**) Relative human mRNA expression for (**B**) *Ki67* (*n* = 6); (**C**) *VEGF*, *HG*F, and *IL8* (*n* = 6); and (**D**) *CXCL1*, *CXCL*5, *CXCL6*, *IL11*, *IL33* and *LIF* (*n* ≥ 3) in infarcted mouse myocardia at day 3 post-engraftment of cardiac mitochondria-preconditioned MSCs in reference to hearts injected with non-treated MSCs. Unpaired Student’s *t*-test in (**A**–**E**) * *p <* 0.05, ** *p <* 0.01, *** *p <* 0.001. Each dot represents a mouse. Bar graphs represent mean values ± SD.

**Figure 6 cells-12-00582-f006:**
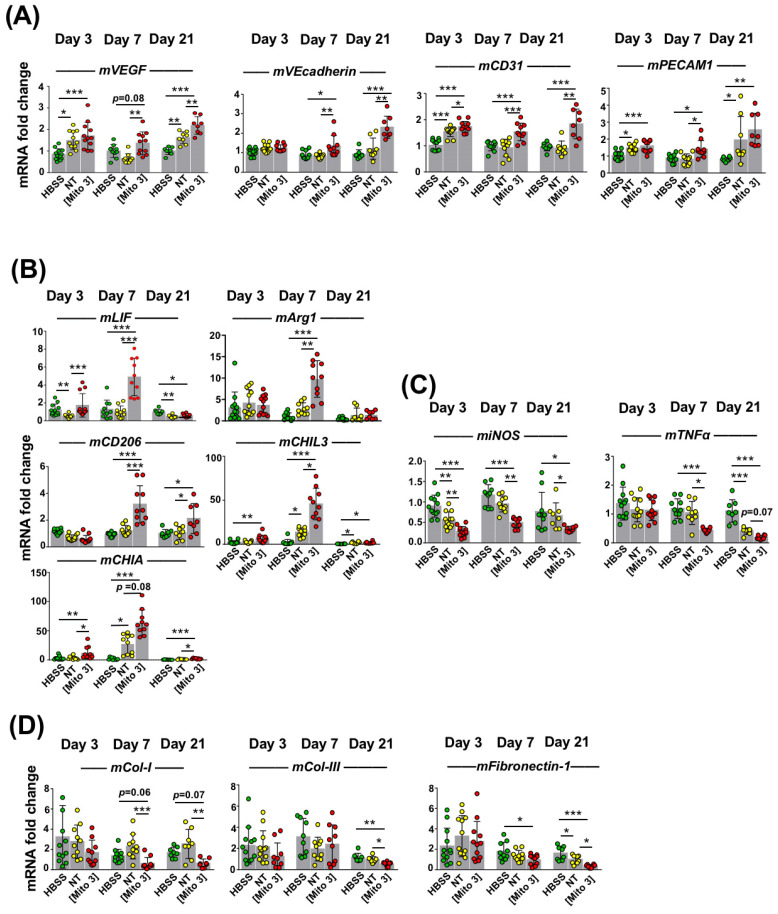
Cardiac mitochondria transfer improves the pro-angiogenic, anti-inflammatory and anti-fibrotic effects of grafted MSCs. (**A**–**D**) MSCs were exposed to cardiac mitochondria at the Mito 3 concentration prior to engraftment in mouse hearts. Relative mouse mRNA expression of (**A**) *VEGF*, *VEcadherin* and *CD31*; (**B**) the anti-inflammatory cytokine *LIF* or the anti-inflammatory M2 macrophage markers (*Arg1*, *CD206*, *CHIL3* and *CHIA*); (**C**) the pro-inflammatory cell marker *iNOS* and the pro-inflammatory cytokine *TNFα*; and (**D**) the extracellular matrix components *collagen-1* (*Col-1*), *collagen-3* (*Col-3)* and *fibronectin-1* in infarcted mouse hearts grafted with either human non-treated or mitochondria-preconditioned MSCs in reference to mouse infarcts injected with a saline solution (HBSS) at day 3 (*n* = 12), day 7 (*n* = 10), and day 21 (*n* = 8) post-surgery and graft. One-way ANOVA with Tukey’s multiple comparisons test for *mVEGF*, *mVEcadherin, mCD31*, *miNOS*, *mTNFα* and *mFibronectin-1*. One-way ANOVA with Dunn’s multiple comparisons tests were performed for the other genes. * *p <* 0.05, ** *p <* 0.01, *** *p <* 0.001. Each dot represents a mouse. Bar graphs represent mean values ± SD.

**Figure 7 cells-12-00582-f007:**
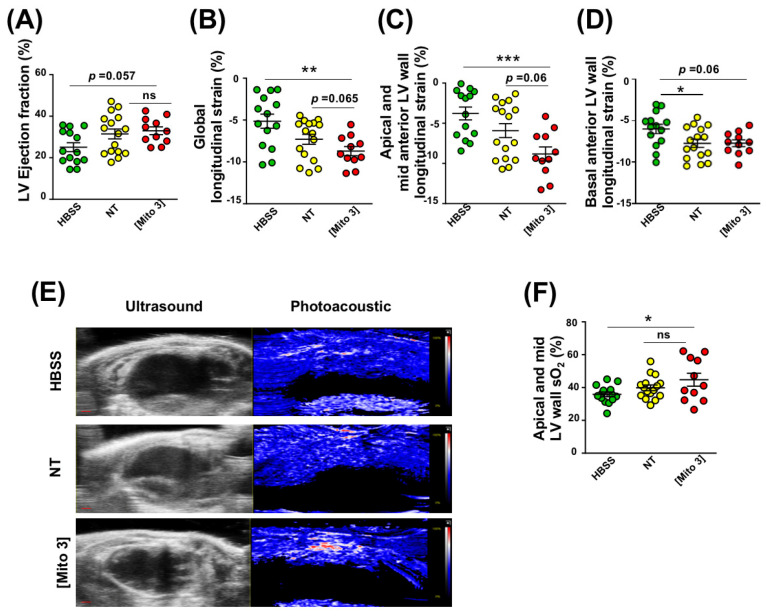
Cardiac mitochondria transfer enhances cardiac function and myocardial perfusion 21 days after myocardial infarction. (**A**–**D**) MSCs were conditioned with cardiac mitochondria (Mito3 concentration) 24 h prior to engraftment in infarcted mouse hearts. Different cardiac functional parameters were evaluated in infarcted mice treated with a saline solution (HBSS), non-treated MSCs (NT), or cardiac mitochondria-preconditioned MSCs (Mito 3) at day 21 post-surgery and graft. (**A**) LV ejection fraction; (**B**) global longitudinal strain; (**C**) apical and mid anterior LV wall longitudinal strain; (**D**) basal anterior LV wall longitudinal strain. (**E**) Representative B-mode ultrasound parasternal long-axis view to define the left ventricular anterior wall and photoacoustic mode with color scaling to show areas of high oxygen saturation in red and low oxygen saturation in blue in hearts from the different mouse treatment groups. Scale bar: 1 mm. (**E**) Variation of myocardial anterior wall sO_2_ in the different mouse treatment groups. One-way ANOVA with Tukey’s multiple comparisons test in (**A**–**D**,**F**). * *p <* 0.05, ***p* < 0.01, *** *p <* 0.001. Each dot represents a mouse. Infarcted mice treated with a saline solution (HBSS) (*n* = 11), non-treated MSCs (NT) (*n* = 11), or cardiac mitochondria-preconditioned MSCs (*n* = 10). Bar graphs represent mean values ± SEM.

## Data Availability

The data presented in this study are available on request from the corresponding author.

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
