# Peer review of "Transfer of Cardiac Mitochondria Improves the Therapeutic Efficacy of Mesenchymal Stem Cells in a Preclinical Model of Ischemic Heart Disease"

_cells, 2023, doi:10.3390/cells12040582_

Round 1
Reviewer 1 Report
Previous studies showed that mitochondria undergo spontaneous transfer from injured cells to MSCs activating cytoprotective and pro-angiogenic pathways in recipient MSCs. The authors studied if the preconditioning of MSCs with mitochondria isolated from human fetal myocytes improves their therapeutic function in ischemic heart disease. They showed that cardiac mitochondria are internalized by MSCs through dynamin-dependent clathrin-mediated endocytosis. They showed that preconditioning stimulates the proliferation and repair functions of MSCs. This effect occurs through the production of ROS. The results of this study indicate that the preconditioning of MSCs with mitochondria may be a novel strategy to improve MSC-based therapy in ischemic heart disease.
This is a very good manuscript with convincing data.
However, in my opinion, the introduction is not broad enough.
One of the findings of this study is the identification of the pathway that allows mitochondria internalization. However, dynamin-dependent clathrin-mediated endocytosis is not the only pathway of mitochondria transfer between cells. Authors should describe the routes of mitochondria intercellular transfer. For example the role of tunneling nanotubes (TNTs) should be discussed. Also how the cells "chose" the route of transfer? Why do some cells transfer mitochondria through TNTs and others by endocytosis? What are the benefits of TNT/versus endocytosis transfer? Can a given cell or cell-type use both pathways? Are these pathways cell or cell-type-specific, or they are interchangable? Adding all this information to the introduction will improve the manuscript.
Author Response
Reviewer 1
Previous studies showed that mitochondria undergo spontaneous transfer from injured cells to MSCs activating cytoprotective and pro-angiogenic pathways in recipient MSCs. The authors studied if the preconditioning of MSCs with mitochondria isolated from human fetal myocytes improves their therapeutic function in ischemic heart disease. They showed that cardiac mitochondria are internalized by MSCs through dynamin-dependent clathrin-mediated endocytosis. They showed that preconditioning stimulates the proliferation and repair functions of MSCs. This effect occurs through the production of ROS. The results of this study indicate that the preconditioning of MSCs with mitochondria may be a novel strategy to improve MSC-based therapy in ischemic heart disease.
This is a very good manuscript with convincing data.
However, in my opinion, the introduction is not broad enough.
One of the findings of this study is the identification of the pathway that allows mitochondria internalization. However, dynamin-dependent clathrin-mediated endocytosis is not the only pathway of mitochondria transfer between cells. Authors should describe the routes of mitochondria intercellular transfer. For example, the role of tunneling nanotubes (TNTs) should be discussed. Also how the cells "chose" the route of transfer? Why do some cells transfer mitochondria through TNTs and others by endocytosis? What are the benefits of TNT/versus endocytosis transfer? Can a given cell or cell-type use both pathways? Are these pathways cell or cell-type-specific, or they are interchangable? Adding all this information to the introduction will improve the manuscript.
We thank Reviewer 1 for his appreciation of the quality of our results and our manuscript.
As recommended, we further developed the questioning about what is known on the routes of mitochondria intercellular transfer, including tunneling nanotubes (TNTs) in the introduction section, as described in the added paragraph, page 2 of the revised version of the manuscript:
”The transfer of mitochondria between cells is now a well-established process that can occur between various cell types. This exchange of mitochondria has often been shown to take place via tunneling nanotubes, which are thin, open-ended tubular structures connecting cells together (see reviews, (12–14)). Alternatively, mitochondrial transfer can also be supported by macropinocytosis (15) or by endocytosis, as it was shown for platelet or macrophage mitochondria entry in MSCs (16,17). What favors one of these means of mitochondrial transfer over the others is still a matter of investigation. However, the outcome of mitochondrial transfer for target cells is increasingly documented.”

Reviewer 2 Report
In the manuscript the authors examined effects of MSCs pre-treated with cardiomyocytes -isolated mitochondria and assessed their therapeutic potential for infarcted mouse hearts . They showed that cardiac mitochondria-preconditioning improves the proliferation and repair properties of MSCs in vitro and in vivo . Mechanistically, cardiac mitochondria produced reactive oxygen species which trigger their degradation in recipient MSCs are playing a key role to activate MSC”s protective function against ischemic heart disease. Overall, this study is well-organized and provides valuable information for readers to understand the impacts of preconditioning MSC with mitochondria on heart repairs following infarction. However there are several issues need to be addressed before consideration for publication.
Major concerns
1) Did authors examine status of mitochondria before incubation with MSC ? Is it possible that damaged mitochondria fragments also provide similar effects on MSC activation?
2) Previous studies indicated MSCs also highly transferred own mitochondria to effectively protect against ischemia heart disease (doi: 10.1016/j.stemcr.2016.08.009), eye diseases (doi: 10.7150/thno.29422 )and lung injury (doi: 10.1165/rcmb.2013-0529OC.) Did authors observe cardiomyocyte-mitochondria preconditioned MSCs also can transfer MSC mitochondria or cardiomyocyte-Mito to ischemic myocardium ? It would be valuable to examine above another direction of MSC mitochondria transfer and discuss this possibility.
3) Quality control and adult MSC. How many batches and what passages of adult MSCs were used in this study ?Although there is no issues about the safety of MSC based therapy for many diseases including heart infarction, a big issue is that there are inconsistent therapeutic efficacy by using adult MSC derived from different tissues including bone marrow, adipose tissues and umbilical cord-tissues etc. A rigorous quality control system of MSC is critical to reduce batch-to-batch variation including secretome . To tackle these challenges, recently MSCs-derived from pluripotent stem cells(iPSC-MSCs) with strong immunomodulation have been established and proposed as an alternative resource to reduce batch-to-batch variation of MSC products(doi: 10.1007/978-1-4939-3584-0_17) . It is worth noting that GMP-grade iPSC-MSCs have been used in refractory graft-versus-host-disease (GVHD) in clinical trials (doi: 10.1038/s41591-020-1050-x) . It would be informative to include above concerns in discussion to enhance understanding quality control for MSC.
Author Response
Reviewer 2
In the manuscript, the authors examined effects of MSCs pre-treated with cardiomyocytes-isolated mitochondria and assessed their therapeutic potential for infarcted mouse hearts. They showed that cardiac mitochondria-preconditioning improves the proliferation and repair properties of MSCs in vitro and in vivo. Mechanistically, cardiac mitochondria produced reactive oxygen species which trigger their degradation in recipient MSCs are playing a key role to activate MSCs protective function against ischemic heart disease. Overall, this study is well-organized and provides valuable information for readers to understand the impacts of preconditioning MSC with mitochondria on heart repairs following infarction. However, there are several issues need to be addressed before consideration for publication.
Major concerns
1) Did authors examine status of mitochondria before incubation with MSC? Is it possible that damaged mitochondria fragments also provide similar effects on MSC activation?
The status of the mitochondria that are isolated from cardiomyocytes for transfer to MSCs is indeed an important issue. We are confident that the phenomena we describe are due to whole mitochondria, and not mitochondria fragments or DAMPS, for several reasons:
1) We have previously shown that mitochondria transferred by platelets or damaged cardiomyocytes/endothelial cells to MSCs have to be respiratory competent and contain mtDNA to activate MSCs (Levoux et al, Cell Metabolism 2021, doi: 10.1016/j.cmet.2020.12.006 ; Mahrouf-Yorgov et al, Cell death and Diff 2017, doi: 10.1038/cdd.2017.51).
2) We showed in the present study that ROS produced by exogenous mitochondria are critical for the activation of MSCs. Since mitochondria fragments fail to produce ROS by themselves, they can’t activate MSCs in the same way as whole mitochondria.
3) Mitochondria fragments signal through interaction with pattern recognition receptors (PRRs) at the cell surface of target cells. Therefore, if these fragments contributed to MSCs activation, such an effect should be detected in MSCs following mitochondria preconditioning in presence of dynasore, that impedes the internalization of mitochondria but not the interaction of DAMPS with their receptors.
This point has been discussed in the novel version of the manuscript, page16 as follows: “Whether isolated mitochondria need to be functional to activate MSCs remains an open question. However, according to our previous data showing that mitochondria transferred by platelets or damaged cardiomyocytes/endothelial cells to MSCs have to be respiratory competent or contain mtDNA to activate MSCs (16,21), it seems that exogenous mitochondria have to be intact to trigger ROS boost in MSCs and therefore, to stimulate their repair properties.”
2) Previous studies indicated MSCs also highly transferred own mitochondria to effectively protect against ischemia heart disease (doi: 10.1016/j.stemcr.2016.08.009), eye diseases (doi: 10.7150/thno.29422) and lung injury (doi: 10.1165/rcmb.2013-0529OC.) Did authors observe cardiomyocyte-mitochondria preconditioned MSCs also can transfer MSC mitochondria or cardiomyocyte-Mito to ischemic myocardium? It would be valuable to examine above another direction of MSC mitochondria transfer and discuss this possibility.
We focused our present study on the effects of mitochondria acquisition by MSCs and did not investigate whether MSCs can transfer their mitochondria following their engraftment into ischemic myocardium. Nevertheless, we previously reported that engrafted MSC transferred their mitochondria to cardiomyocytes in vivo (Figeac et al, Stem cells, 2014). In addition, we also showed in the same study that human MSCs, having received cardiomyocyte mitochondria through coculture with damaged mouse cardiomyocytes, were also able to transfer their mitochondria to cardiac cells following their delivery into ischemic heart. It is worth noting that the preconditioning of MSCs with damaged cells, like the preconditioning with exogenous mitochondria, improves the heart repair capacity of MSCs and enhances their therapeutic secretome. Accordingly, we believe that mitochondria-preconditioning should increase the capacity of engrafted MSCs to donate their mitochondria to surrounding cells in ischemic myocardium, thus contributing to repair damaged heart. In contrast, although this issue remains to be formally addressed, it is unlikely that engrafted preconditioned MSCs transfer cardiac mitochondria to cardiomyocytes in vivo since these foreign mitochondria are shown in the present study to be rapidly degraded.
The potential occurrence and role of mitochondria transfer from MSCs to cardiac resident cells in vivo has been further discussed in the revised version of the manuscript, pages 16 and17, as follows:
“Mitochondria trafficking from engrafted MSCs to surrounding damaged cells has also been shown to have pro-healing effects in injured heart, retina and lung tissues (43–45). In particular, occurrence of this process in vivo has been documented to improve the energetic metabolism and thus the survival of distressed cells. In addition, transfer of MSC mitochondria to cardiomyocytes and renal tubular cells has also reported in vitro to be associated with the differentiation of MSCs into these cell types (20,46).”
….
“Interestingly, following their engraftment into mouse ischemic heart, naive MSCs as well as MSCs that previously received, in vitro, mitochondria from co-cultured cardiomyocytes were found to transfer their mitochondria to cardiomyocytes in vivo (10). These observations suggest that engrafted mitochondria-preconditioned MSCs may also be capable to transfer their mitochondria in vivo to heart resident cells. Nevertheless, further investigations are clearly needed to formally determine the respective contributions for heart repair of mitochondria trafficking and cytokine secretion from the cardiac mitochondria-conditioned MSCs remains to be determined.”
3) Quality control and adult MSC. How many batches and what passages of adult MSCs were used in this study? Although there is no issues about the safety of MSC based therapy for many diseases including heart infarction, a big issue is that there are inconsistent therapeutic efficacy by using adult MSC derived from different tissues including bone marrow, adipose tissues and umbilical cord-tissues etc. A rigorous quality control system of MSC is critical to reduce batch-to-batch variation including secretome. To tackle these challenges, recently MSCs-derived from pluripotent stem cells(iPSC-MSCs) with strong immunomodulation have been established and proposed as an alternative resource to reduce batch-to-batch variation of MSC products (doi: 10.1007/978-1-4939-3584-0_17). It is worth noting that GMP-grade iPSC-MSCs have been used in refractory graft-versus-host-disease (GVHD) in clinical trials (doi: 10.1038/s41591-020-1050-x). It would be informative to include above concerns in discussion to enhance understanding quality control for MSC.
We used hMADS cells isolated from adipose tissues from two donors at passages from 10 to 30 since these cells have strong self-renewal potential due to their telomerase activity (Rodriguez AM et al, J Exp Med 2005, doi: 10.1084/jem.20042224). These informations were added in the material and method section of the new version of the manuscript page 5 as follows:” Experiments were conducted with hMADS cells isolated from two young donors at passages 10 to 30.”
Of note, although not shown in the present study, we confirmed that mitochondria preconditioning exerts similar activation in MSCs isolated from bone marrow of 5 adult donors at passage 2.
We agree that a rigorous quality control system of MSC is critical to reduce batch-to-batch variation including secretome. With this regard, the use of MSCs-derived from pluripotent stem cells(iPSC-MSCs) could be a promising alternative to reduce batch-to-batch variation of MSC products. This critical point for future clinical applications is now discussed in the new version of the manuscript, page 18, as follows:
“Finally, one of the most critical issues raised by our study, and related to clinical applications, concerns the high functional variability of MSCs depending on the age and health status of the donors, thus requiring rigorous MSC quality control. To circumvent this problem, mitochondria preconditioning could be advantageously performed on pluripotent stem cell-derived MSCs (iPSC-MSCs) which have strong immunomodulatory properties, can be produced at clinical grade (53) and are already tested in clinical trials (54).”
Reviewer 3 Report
In general, the work is of great interest. On the other hand, because of the fundamental result, and due to the too great significance of this result, it requires serious evidence. The critical question is whether donor mitochondria penetrate into the cell or the entire protective signaling is provided only by interaction with the cell surface. Although the authors claim the first, it is necessary to give more arguments in support of this.
It is clear that the main research tool was PCR data on the analysis of MSCs RNA under different experimental conditions, and all other approaches supporting the conclusions have less expertise from the authors. There are especially a lot of questions around the details of Figure 1. Firstly, the quality of confocal images does not allow for a clear examination. There is no transmittance image, which made it possible not to admit that cells with stained mitochondria were not added together with the mitochondria. To prevent this, it is necessary to stain the system with a cellular marker of cardiomyocytes (e.g., β-myosin H-chain).
In addition, it was not explained what the stain WGA (red channel) is, and in general, why it is needed, and if this dye stains the cytoplasm, why donor mitochondria do not fall into all zones of the cytosol, while there is a large blind and inaccessible space for the introduction of mitochondria.
Confocal microscopy allows to determine the localization of fluorescent profiles inside the object and scanning along the z-axe would allow to determine whether a fluorescent object is outside or inside the cell, what needs to be done.
Also. it raises the question whether MTG could not be released from the donor mitochondria into the medium during 24 hours of incubation and then stain the host mitochondria. It is unclear how well the mitochondria were washed after loading with MTG (this, by the way, also has to do with loading with TEMPO). There are several ways to solve this issue, in particular, you can incubate the mitochondria with MTG separately for 24 hours, then centrifuge and add a supernatant to the cells. Or you can stain the MSCs mitochondria with another dye that fluoresces in a different than MTG spectral range. It is also possible to transfect MSCs with a fluorescent protein with other fluorescent characteristics and visualize the host mitochondria, etc.
For some reason, in Figure 1, flow cytometry does not record the red channel in WGA cells staining after the addition of mitochondria.
There is another point that requires analysis. The authors discuss only one of the mechanisms that accompanies the transfer of mitochondria, in which MSCs participate, namely, when the transfer of mitochondria FROM differentiated cells TO MSCs improves the properties of MSCs. However, there is a whole layer of pioneering works in which it is shown that unidirectional transfer of mitochondria FROM MSCs TO differentiated cells leads to changes in MSCs, which in particular are associated with subsequent differentiation of MSCs (e.g., doi: 10.1111/j.1582-4934.2007.00205.x. , doi: 10.1016/j.yexcr.2010.06.009, doi: 10.5966/sctm.2015-0010). This diversity of multidirectional mitochondrial transport FROM and TO the MSC should definitely be discussed.
Author Response
Reviewer 3
- In general, the work is of great interest. On the other hand, because of the fundamental result, and due to the too great significance of this result, it requires serious evidence. The critical question is whether donor mitochondria penetrate into the cell or the entire protective signaling is provided only by interaction with the cell surface. Although the authors claim the first, it is necessary to give more arguments in support of this.
We thank Reviewer 3 for appreciating the quality of our work and its important implications.
Different works, including ours, show that mitochondria can be transferred between cells via TNTs, microvesicles or in free form. Our previous work (Levoux et al, Cell Metabolism, 2021) as well as that of Cai W (Advance Science 2022) (https://pubmed.ncbi.nlm.nih.gov/36507570/) indicate that mitochondria in free form as well as those contained in microvesicles are internalised by MSCs via a clathrin-mediated dynamin-dependent endocytosis mechanism. In the present study, our results indicate that mitochondria previously isolated from cardiomyocytes are also predominantly internalized by MSCs via this same endocytosis modality.
Although we cannot exclude that isolated mitochondria may activate MSCs by interacting with the cell surface, the contribution of this mechanism remains minimal because in the presence of dynasore, which inhibits internalization of exogenous mitochondria into MSCs, preconditioning with exogenous mitochondria does not activate the regenerative properties of MSCs.
To make this point clearer, the manuscript has been modified accordingly page 15 as follows: “Although we cannot exclude that isolated mitochondria may activate MSCs by interacting with the cell surface, the contribution of this mechanism remains minimal because in the presence of dynasore, which inhibits internalization of exogenous mitochondria into MSCs, preconditioning with exogenous mitochondria does not activate the regenerative properties of MSCs.” in the discussion section.
- It is clear that the main research tool was PCR data on the analysis of MSCs RNA under different experimental conditions, and all other approaches supporting the conclusions have less expertise from the authors. There are especially a lot of questions around the details of Figure 1. Firstly, the quality of confocal images does not allow for a clear examination. There is no transmittance image, which made it possible not to admit that cells with stained mitochondria were not added together with the mitochondria. To prevent this, it is necessary to stain the system with a cellular marker of cardiomyocytes (e.g., β-myosin H-chain).
Response
It is highly unlikely that cells can survive the mitochondria isolation procedure. Nevertheless, this is an important issue that needs to be addressed adequately. With that purpose, we incubated the cardiac mitochondria preparation under cardiomyocyte culture conditions. In these conditions, no cardiac cells were found to grow after several days of culture, thus confirming the lack of living cells in the mitochondria suspension. This information is now provided in the manuscript, in the Methods section, page 5, as follows: “To check the absence of living cells in the mitochondria suspension, cardiomyocyte mitochondria were incubated under cardiomyocyte culture conditions. In these conditions, no cardiac cells we found to grow after several days of culture“.
- In addition, it was not explained what the stain WGA (red channel) is, and in general, why it is needed, and if this dye stains the cytoplasm, why donor mitochondria do not fall into all zones of the cytosol, while there is a large blind and inaccessible space for the introduction of mitochondria.
Confocal microscopy allows to determine the localization of fluorescent profiles inside the object and scanning along the z-axe would allow to determine whether a fluorescent object is outside or inside the cell, what needs to be done.
WGA is a probe for detecting glycoconjugates, which selectively bind to N-acetylglucosamine and N-acetylneuraminic acid residues of cell membranes. WGA conjugated to Alexa Fluor dyes is widely used to stain mammalian cell membranes and perform colocalization studies, in particular involving organelles. Therefore, this WGA probe has been extensively used in publications on intercellular mitochondria transfer, in particular to visualize tunneling nanotubes mediating these transfers (to cite a few of these articles: Walters HE et al, Oxid Med Cell Longev. 2021, doi: 10.1155/2021/6697861; van der Vlist M, et al, Neuron. 2022 doi: 10.1016/j.neuron.2021.11.020; Wang X and Gerdes HH, Cell Death Differ 2015 https://doi.org/10.1038/cdd.2014.211, Pasquier J et al., J Transl Med 2013 https://doi.org/10.1186/1479-5876-11-94). Intercellular transfer of mitochondria was detected by confocal microscopy/time-lapse imaging in the WGA-stained cells (Wang X and Gerdes HH, Cell Death Differ 2015 https://doi.org/10.1038/cdd.2014.211, Pasquier J et al., J Transl Med 2013 https://doi.org/10.1186/1479-5876-11-94).
In our present study, we used WGA to facilitate the detection of MSCs in confocal microscopy imaging. In order not to interfere with mitochondria acquisition, MSCs were stained with WGA after their preconditioning with cardiac mitochondria and just before confocal microscopy analysis.
This information has been added in the material and method of the revised version of the manuscript page 5, as follows: “Just before confocal microscopy analysis, MSCs were incubated with wheat germ agglutinin (WGA) conjugated to Alexa Fluor 647 (5 μg/mL; Invitrogen, Cat#W32466) after their exposure to isolated MitoTracker-stained mitochondria to facilitate their detection.”
We fully agree that z-axis scanning and 3D reconstruction allows precise localization of objects. This is why we performed such analyses in previous publications, supporting the intracellular localization of transferred MSC mitochondria inside MDA-MB231 cancer cells and glioblastoma cells (Caicedo et al., 2015 doi: 10.1038/srep09073; Nzigou Mombo et al., 2017 doi: 10.3791/55245) and in human and murine T cells (Luz-Crawford et al., 2019 doi: 10.1186/s13287-019-1307-9; Akhter et al., 2023 doi: 10.1186/s13287-022-03219-x).
We did not perform here Z-stack acquisitions. Instead, we are showing 350 nm-thick cell slice (Airyscan imaging). The large black areas correspond to the positioning of nuclei, which can be identified by phase contrast. The transferred cardiac mitochondria display typical networks within the cytoplasm and around the nucleus.
We added the following sentence, page 5, in the result section of the revised manuscript: “After 24 hour-exposure, and consistent with our earlier observations, confocal microscopy showed that MSCs contained cardiac mitochondria with a typical network distribution around their nucleus (Figure 1A).”
- Also. it raises the question whether MTG could not be released from the donor mitochondria into the medium during 24 hours of incubation and then stain the host mitochondria. It is unclear how well the mitochondria were washed after loading with MTG (this, by the way, also has to do with loading with TEMPO). There are several ways to solve this issue, in particular, you can incubate the mitochondria with MTG separately for 24 hours, then centrifuge and add a supernatant to the cells. Or you can stain the MSCs mitochondria with another dye that fluoresces in a different than MTG spectral range. It is also possible to transfect MSCs with a fluorescent protein with other fluorescent characteristics and visualize the host mitochondria, etc.
We fully agree that MitoTraker leakage following staining is a critical issue that needs to be strictly monitored to assess and limit its extent. With this goal, MitoTraker-stained cardiac cells are extensively rinsed, immediately after the staining and, 24h later, before mitochondria isolation, which itself includes several centrifugation steps. Consistent with the absence of MSC mitochondria staining due to MitoTraker leakage from MitoTraker-stained cardiac mitochondria, no MitoTraker fluorescence was observed in MSCs treated with dynasore, as measured both by confocal microscopy or flow cytometry.
The protocol of exogenous cardiac mitochondria preconditioning of MSCs is now further detailed in the Methods section of the new version of the manuscript, page 5, as follows: “To avoid MitoTracker leakage from stained cardiac mitochondria (and parasite staining of MSC mitochondria), cardiac cells were washed 3 times with Phosphate Buffered Saline solution (PBS, Thermo Scientific Cat #10010023) immediately after MitoTracker staining, and again 3 times 24h later, before mitochondria isolation.”
- For some reason, in Figure 1, flow cytometry does not record the red channel in WGA cells staining after the addition of mitochondria.
For flow cytometry analysis, we did not stain MSCs with WGA because Forward scatter and Size scatter parameters in these conditions were sufficient to identify the MSC population.
- There is another point that requires analysis. The authors discuss only one of the mechanisms that accompanies the transfer of mitochondria, in which MSCs participate, namely, when the transfer of mitochondria FROM differentiated cells TO MSCs improves the properties of MSCs. However, there is a whole layer of pioneering works in which it is shown that unidirectional transfer of mitochondria FROM MSCs TO differentiated cells leads to changes in MSCs, which in particular are associated with subsequent differentiation of MSCs (e.g., doi: 10.1111/j.1582-4934.2007.00205.x. , doi: 10.1016/j.yexcr.2010.06.009, doi: 10.5966/sctm.2015-0010). This diversity of multidirectional mitochondrial transport FROM and TO the MSC should definitely be discussed.
We agree that our present manuscript focuses on the mechanisms that accompany the transfer of mitochondria, from differentiated cells to MSCs and shows how this transfer improves the properties of MSCs. However, in vivo, it is possible that engrafted MSCs release mitochondria to cardiac resident cells thus contributing to their activation or to ischemic heart repair.
This diversity of multidirectional mitochondrial transport from and to the MSCs is now discussed in the discussion section of the revised form of the manuscript pages 16 and 17, as follows:
“Mitochondria trafficking from engrafted MSCs to surrounding damaged cells has also been shown to have pro-healing effects in injured heart, retina and lung tissues (43–45). In particular, occurrence of this process in vivo has been documented to improve the energetic metabolism and thus the survival of distressed cells. In addition, transfer of MSC mitochondria to cardiomyocytes and renal tubular cells has also reported in vitro to be associated with the differentiation of MSCs into these cell types (20,46).”
….
“Interestingly, following their engraftment into mouse ischemic heart, naive MSCs as well as MSCs that previously received, in vitro, mitochondria from co-cultured cardiomyocytes were found to transfer their mitochondria to cardiomyocytes in vivo (10). These observations suggest that engrafted mitochondria-preconditioned MSCs may also be capable to transfer their mitochondria in vivo to heart resident cells. Nevertheless, further investigations are clearly needed to formally determine the respective contributions for heart repair of mitochondria trafficking and cytokine secretion from the cardiac mitochondria-conditioned MSCs remains to be determined.”

Round 2
Reviewer 3 Report
I think that the authors properly addressed my critique thus resulting in improved version which can be accepted